# Heterogeneity of Phase II Enzyme Ligands on Controlling the Progression of Human Gastric Cancer Organoids as Stem Cell Therapy Model

**DOI:** 10.3390/ijms242115911

**Published:** 2023-11-02

**Authors:** Deng-Chyang Wu, Chia-Chen Ku, Jia-Bin Pan, Kenly Wuputra, Ya-Han Yang, Chung-Jung Liu, Yi-Chang Liu, Kohsuke Kato, Shigeo Saito, Ying-Chu Lin, Inn-Wen Chong, Michael Hsiao, Huang-Ming Hu, Chao-Hung Kuo, Kung-Kai Kuo, Chang-Shen Lin, Kazunari K. Yokoyama

**Affiliations:** 1Graduate Institute of Medicine, Kaohsiung Medical University, Kaohsiung 807, Taiwan; dechwu@yahoo.com (D.-C.W.); r991046@gap.kmu.edu.tw (C.-C.K.); r060139@gap.kmu.edu.tw (J.-B.P.); kenlywu@hotmail.com (K.W.); chong@kmu.edu.tw (I.-W.C.); csl@kmu.edu.tw (C.-S.L.); 2Regenerative Medicine and Cell Therapy Research Center, Kaohsiung Medical University, Kaohsiung 807, Taiwan; sal9522059@yahoo.com.tw (Y.-H.Y.); pinkporkkimo@yahoo.com.tw (C.-J.L.); kkkuo@kmu.edu.tw (K.-K.K.); 3Cell Therapy and Research Center, Kaohsiung Medical University Hospital, Kaohsiung 807, Taiwan; ycliu@kmu.edu.tw; 4Division of General and Digestive Surgery, Department of Surgery, Kaohsiung Medical University Hospital, Kaohsiung 807, Taiwankjh88@kmu.edu.tw (C.-H.K.); 5Division of Gastroenterology, Department of Internal Medicine, Kaohsiung Medical University Hospital, Kaohsiung 807, Taiwan; 6Department of Infection Biology, Graduate School of Comprehensive Human Sciences, The University of Tsukuba, Tsukuba 305-8577, Japan; kkato@md.tsukuba.ac.jp; 7Saito Laboratory of Cell Technology, Yaita 239-1571, Japan; saict1@maple.ocn.ne.jp; 8School of Dentistry, Kaohsiung Medical University, Kaohsiung 807, Taiwan; chulin@cc.kmu.edu.tw; 9Division of Pulmonary and Critical Care Medicine, Kaohsiung Medical University Hospital, Kaohsiung 807, Taiwan; 10Department of Biological Science and Technology, National Yang Ming Chiao Tung University, Hsinchu 300, Taiwan; 11Genome Research Center, Academia Sinica, Nangan, Taipei 115, Taiwan; mhisao@gate.sinica.edu.tw; 12Department of Internal Medicine, Kaohsiung Municipal Ta-Tung Hospital, Kaohsiung 801, Taiwan; 13Department of Internal Medicine, Kaohsiung Municipal Siaogang Hospital, Kaohsiung 812, Taiwan

**Keywords:** cinnamaldehyde, gastric cancer development, organoids, perillaldehyde, reactive oxygen species

## Abstract

Gastric cancer (GC) organoids are frequently used to examine cell proliferation and death as well as cancer development. Invasion/migration assay, xenotransplantation, and reactive oxygen species (ROS) production were used to examine the effects of antioxidant drugs, including perillaldehyde (PEA), cinnamaldehyde (CA), and sulforaphane (SFN), on GC. PEA and CA repressed the proliferation of human GC organoids, whereas SFN enhanced it. Caspase 3 activities were also repressed on treatment with PEA and CA. Furthermore, the tumor formation and invasive activities were repressed on treatment with PEA and CA, whereas they were enhanced on treatment with SFN. These results in three-dimensional (3D)-GC organoids showed the different cancer development of phase II enzyme ligands in 2D-GC cells. ROS production and the expression of TP53, nuclear factor erythroid 2-related factor (NRF2), and Jun dimerization protein 2 were also downregulated on treatment with PEA and CA, but not SFN. NRF2 knockdown reversed the effects of these antioxidant drugs on the invasive activities of the 3D-GC organoids. Moreover, ROS production was also inhibited by treatment with PEA and CA, but not SFN. Thus, NRF2 plays a key role in the differential effects of these antioxidant drugs on cancer progression in 3D-GC organoids. PEA and CA can potentially be new antitumorigenic therapeutics for GC.

## 1. Introduction

Gastric cancer (GC) is known as the fifth most malignant cancer and the second leading cause of cancer-related death in both sexes worldwide [1]. In Taiwan, it was the sixth leading cause of cancer-related death in 2018 [2]. Furthermore, histological subtyping of GC showed that more than 90% of patients with GC had adenocarcinomas. The Cancer Genome Atlas classified gastric adenocarcinoma into four molecular subgroups [3]. One GC group is positive for the Epstein-Bar virus (EBV) with frequent mutations of the phosphatidylinositol-4,5-bisphosphate 3-kinase catalytic subunit alpha gene and silencing of the cyclin-dependent kinase inhibitor 2A gene. A microsatellite instable subtype has a hypermutation phenotype. A genomic stable subtype displays a diffuse type of histology and frequent mutations of the cadherin and Ras *RAS* homolog family member A genes. A chromosomal instable and Ras *RAS* homolog family member A subtype caused the aneuploidy and frequent mutation of TP53 as well as activation of the receptor tyrosine kinase-RAS pathway. The prognosis of GC is poor. Moreover, the lack of clinical signs leads to a delay in the diagnosis, with 75% of patients presenting with incurable advanced disease [4]. In general, surgery is the only curative option. Furthermore, chemotherapy is thought to improve survival rates [5]. The most used chemotherapeutics for GC is fluoropyrimidines, including 5-fluorouracil and capecitabine, and platinum compounds, such as cisplatin, oxaliplatin, and docetaxel [5,6]. Besides classical chemotherapy, novel therapies targeting genetic alterations present another treatment option. Until now, the only approved therapies are trastuzumab to target the epidermal growth factor receptor-2 signaling and ramucirumab to target the vascular endothelial growth factor axis [7,8]. Treatments using the anti-epidermal growth factor receptor antibodies cetuximab and panitumumab and the anti-vascular endothelial growth factor antibody bevacizumab have failed to provide good survival rates. One of the reasons for their poor efficacy is the lack of identification of efficient biomarkers to direct these targeted therapies to the appropriate patients.

The three-dimensional (3D) culture systems called organoids might present a new opportunity in preclinical personalized therapy. These culture systems recapitulate many aspects of the tissues and organs as well as the microenvironments faithfully [9,10,11]. The advantages of organoid cultures are as follows: results can be obtained more rapidly using organoid assays than xenotransplantation assays, and it is possible to prepare large organoid samples from the same patients. For drug development and screening, organoids can be used instead of performing drug treatment for each patient [12]. For example, the merits of organoids have been reported in the colorectal cancer biobank, which is amenable to drug screening for individual patient sample treatment and developing novel therapeutics [13]. In addition to the primary cancer-derived organoids, gastrointestinal organoids from metastatic lesions can also be prepared to study the patient’s clinical response [14,15].

Perillaldehyde (PEA) is extracted from *Perilla frutescens* [16] and is widely used as a constituent of essential oils and in Asian cuisine. It has strong antifungal, anti-inflammatory, antitumor, and antioxidant activities as well as many other biological activities [17,18,19]. Recently, Furuno et al. reported that PEA activated the nuclear factor erythroid 2-related factor (NRF2)/heme oxygenase-1 (HO-1) pathway and reduced the oxidative stress-mediated innate immune response in human keratinocytes [20].

Cinnamaldehyde (CA) is a β-unsaturated aldehyde, which is rich in cinnamon and widely used as a food additive and in Chinese medicine [21]. Moreover, the antioxidant activity and anticerebral thrombotic ability of CA have been proven in mice [22]. A recent study indicated that CA induced autophagy-mediated cell death through endoplasmic reticulum stress and epigenetic modification by G9a [23]. Uchi et al. reported that CA inhibited aryl hydrocarbon (AHR) receptor signaling and induced NRF2-mediated antioxidant activity [24]. PEA and CA might be involved in the NRF2 dependent antioxidation reactions to block reactive oxygen species (ROS) production without affecting the AHR-mediated oxidative stress reactions.

Sulforaphane (4-methylsulfinybutyl isothiocyanate; SFN) is a dietary isothiocyanate synthesized from a precursor found in cruciferous vegetables of the genus *Brassica*. It is one of the most representative inducers of the phase II enzyme systems, such as nicotinamide adenine dinucleotide phosphate quinone dehydrogenase 1 (NQO1), glutathione S-transferase alpha 1, and HO-1, which are responsible for eliminating chemicals that damage DNA. SFN is known to induce S-phase cell cycle arrest and apoptosis in a TP53-dependent manner in GC cells [25]. In vitro studies demonstrated that SFN regulates the actin cytoskeleton, inhibiting the formation of actin stress fibers and the expression of associated proteins that promote the spread of premalignant cells. Consequently, SFN suppresses breast cancer metastasis [26].

Another important aspect is the differential regulation of AHR and NRF2 transcription factor-mediated response of phase I and II genes. PEA and CA activated the NRF2-mediated phase II response through antioxidant response element (ARE)-mediated promoter activation [20] but inhibited phase I response by AHR activation [24]. However, SFN induced not only phase II promoters by activation of NRF2 transcription factor, but also phase I AHR-mediated phase II target genes like NQO1 [27,28]. Thus, PEA or CA and SFN exhibited to activate the different target genes.

The present study investigated the effects of antioxidant drugs on organoids on a chip, including PEA, CA, and SFN, on the development of GC-derived organoids. Both PEA and CA inhibited GC development and invasive activities by decreasing ROS production and apoptosis through NRF2. However, SFN did not show such effects. Thus, these antioxidant drugs mediate different effects on GC development. Moreover, both PEA and CA can potentially be new antitumorigenic therapeutics for GC.

## 2. Results

### 2.1. Characterization of Antrum Organoids of Patients with GC

We previously generated organoids from the antrum of patients with GC as described elsewhere [11,29,30] to treat them with various reagents, such as the rho-associated protein kinase inhibitor Y-27632, activin A, bone morphogenetic protein, glycogen synthase kinase-3 inhibitor CHIR99021, fibroblast growth factor 4, noggin, retinoic acid, and epidermal growth factor. However, this culture condition required approximately 4 weeks to generate antrum- or corpus-derived organoids [31,32]. Thus, to save time in the present study, we used the bioreactor to replace the flat-stay culture condition [30,32,33]. After 7 days, we could generate organoids of HCM-BROD-0045-C16 that developed two epithelial layers with a central cavity (Figure 1a). We found three types of organoids, including cystic organoids with or without a multilayered wall, organoids with a non-coherent grape-like growth pattern, and organoids with a compact cell cluster and no lumen [13]. These different morphologies were grouped into three subtypes, which were correlated with the Lauren classification [34]. The first subtype is a solid subtype derived from diffuse GC showing amorphous solid configurations and a dis-cohesive growth pattern; the second is a glandular subtype derived from intestinal cancer showing glandular structures with a single lumen; and the third is the mixed subtype [35,36].

The bioreactor-prepared organoids from HCM-BROD-0045-C16 were stained using the muscle biomarker α-SMA (Figure 1b) and the epithelial surface protein marker integrin α6 (Figure 1c). Next, the tumor formation capability was examined using xenotransplantation assays [12,13,37]. The Matrigel-embedded organoids were generated and injected as xenografts into the skin of severe combined immunodeficiency (SCID) male mice. The xenografts were implanted for 8–12 weeks [38,39]. Tumors developed (Figure 1d), and they were stained using antibodies against α-SMA and integrin α6 (Figure 1e,f), which showed a histochemical specimen similar to those of the injected organoids.

### 2.2. Effect of Antioxidant Drugs on Organoid Growth

The effects of antioxidant drugs, including PEA, CA, and SFN, on the growth of the established organoids were investigated using size–mass and viability luminescence assays. The exposure to these drugs for 1 to 28 days was compared (Appendix A). For example, the organoids (~500 µm per well) were cultivated in the grow medium and we investigated the growth and their morphologies on day 1, 7, 14, 21, and 28. We have examined the effects on the series of antioxidation drugs including PEA, CA, tert-Butyl hydroquinone (tBHQ), SFN, N-Acetyl cysteine, DMSO, and control Dulbecco phosphate balanced saline (DPBS), pH 7.4 on the growth of 3D organoids. Here, specifically, we focused upon the PEA, CA, and SFN. The morphology of the control organoids showed uneven and nonrounded clusters as well as distorted aggregates in each case (see the control in Figure 2a,d,g). By contrast, 10 μM PEA treatment decreased the growth of the organoids by 74–87% compared with that in the nontreatment condition (Figure 2a–c). In this case, the morphology of the organoids was round and smooth-round clustered (Figure 2a). In the case of 1 μM CA treatment, similar growth delay (decreased by 72–85%) and uneven surface and distorted structures were detected (Figure 2d–f). By contrast, 10 μM SFN treatment increased the growth of the organoids by 1.2-fold to 1.3-fold compared with that in the nontreatment condition (Figure 2g–i). Distorted and nonrounded structures with higher viability luminescence were detected significantly more frequently in the SFN-treated organoids than in the control organoids, which were different from those in the case of PEA and CA treatments (Figure 2c,f,i).

### 2.3. Effect of Antioxidant drugs on Invasive Activities

Next, the invasive activities of the organoids treated with PEA, CA, and SFN were examined. The original methods for two-dimensional (2D) transformed cells were modified to use for 3D organoids based on previously described protocol [35,36].

The organoids treated with PEA, CA, and SFN were added to the invasion chambers and incubated for 7 days. After harvesting, they were stained using hematoxylin (HE) to determine the invasive activity of each organoid. As shown in Figure 3, PEA and CA treatments decreased the invasive activities by approximately 40–55% compared with those in untreated organoids (Figure 3a,b). By contrast, SFN treatment increased invasive activities by 1.9-fold to 2.1-fold compared with those in control, untreated organoids (Figure 3c).

### 2.4. PEA and CA Repress the Organoid-Mediated Tumor Formation, but SFN Increases Tumor Development

Then, the tumor formation activities of the organoids treated with PEA, CA, and SFN were examined. Comparative studies using xenotransplantation assays were used. As shown in Figure 4a, PEA and CA treatments resulted in the shrinkage of tumors and the masses of xenografts after 8 weeks by approximately 15–35%. However, SFN treatment caused significant increases (approximately 3- to 3.5-fold) in the tumor size and mass compared with those in the control.

The histochemical studies using the antibodies against NRF2, AHR, and p21^Cip1^ revealed that PEA and CA treatments caused a significant reduction in the NRF2, p21^Cip1^, and AHR staining, but SFN treatment enhanced the expressions of these markers (Figure 4b). The control α-SMA staining showed similar profiles. Thus, PEA and CA treatments significantly reduced tumor development, which was completely different from the effect of SFN treatment.

### 2.5. Expression of Stress-Related Factors and TP53

The tumor suppressive effects of PEA and CA were examined further to compare the expression of TP53, NRF2, p21^Cip1^, JDP2, and AHR, which are related to oxidative stress or antioxidation-related reactions, using Western blotting (Figure 4c). The expressions of TP53, NRF2, p21^Cip1^, and JDP2 significantly decreased by 15 to 45% in organoids treated with PEA and CA compared with that in the control organoids. However, the expressions of NRF2 and TP53 in organoids treated with SFN increased about 1.50- to 1.75-fold in the control organoids. The expression of AHR was not changed dramatically about 80% to 110% as compared with the organoids as control. Thus, treatment with PEA and CA reduced the expressions of proteins of the TP53–NRF2 axis including JDP2 and p21^Cip1^, but the SFN increased expression of TP53 and NRF2.

To detect the functional TP53 and MDM2 interaction in cell cycle arrest, we used nutlin-3, which inhibits this interaction [40]. Cancer cells with mutated TP53 are not affected by this compound. As expected, tumor organoids were insensitive to nutlin-3 treatment, indicating a mutated TP53 pathway in these organoids [41] (Figure 4d). In fact, we isolated the mutation of TP53 cDNA at the DNA binding site.

### 2.6. PEA and CA Decrease ROS and Caspase 3/7 Activities, but SFN Increases Them

ROS production in the 3D organoids on treatment with PEA, CA, and SFN was examined as described in the Materials and Methods. On treatment with 10 μM of PEA and 1 μM CA, ROS production decreased by 30% and 40%, respectively, compared with that in the control organoids. However, 10 μM SFN treatment increased ROS production by 1.1-fold compared with that in untreated organoids (Figure 5a). Caspase 3/7 activities were also tested on treatment with PEA, CA, and SFN. PEA and CA treatments reduced caspase 3/7 activities by 70% and 55%, respectively, compared with those in the untreated organoids. However, SFN treatment increased caspase 3/7 activities by 1.25-fold compared with those in the untreated organoids (Figure 5b).

### 2.7. Effects of shRNA against NRF2 on Invasive Activities in Response to PEA, CA, and SFN

To confirm that the antioxidant effects of these drugs are mediated by the NRF2 transcription factor activity, we constructed and introduced *shNRF2* (Figure 6a,b) and then examined the effects on cell invasion after PEA, CA, and SFN treatments (Figure 6c). Treatment with *shNRF2* decreased the expression of NRF2 by 21–26% (Figure 6b). On *shNRF2*-mediated knockdown, the invasive activities in response to PEA and CA treatments increased significantly by 4.2-fold and 3.7-fold, respectively, compared with those in control organoids. However, the SFN-induced invasive activity decreased by 52% compared with that in control organoids (Figure 6d). Thus, *shNRF2* inhibited the PEA and CA induced the invasive activities but not in the case of SFN. These findings confirmed that the differences in the invasive activities might be due to the expression of the NRF2 transcription factor.

## 3. Discussion

The present study showed that antioxidant drugs, such as PEA and CA, prevented the progression of cancer invasion and the development of human GC-derived organoids. By contrast, the antioxidant drug SFN increased cancer progression. Even for the same category of antioxidant drugs, we demonstrated heterogeneous effects on GC development using organoids. This might be due to the incomplete activation of NRF2-driven master genes for cancer development. We also developed a rapid preparation technique for organoids to save time spent growing them. In general, the preparation of human antrum GC organoids requires more than 4 weeks under the regular culture condition [29,35,36]. The long duration required for organoid generation is a problem for the research community. Thus, we used the bioreactor method to decrease the time required for organoid generation (approximately 1 week). This method was validated using morphological immunostaining and the evaluation of the tumor formation capability of organoid xenografts. α-SMA is a marker for a subset of activated fibro-genic cells, called myofibroblasts, which are important effector cells of tissue fibrogenesis [42]. Integrin α6 has been shown to affect the survival of breast carcinoma cells in association with integrin β4 [43]. Both markers exhibited similar expressions in the gastric organoids between the stationery and bioreactor culture conditions. The transformation activities were also maintained (Figure 1).

Initially, we examined the tumor formation ability of organoids derived from human-induced pluripotent stem cell clones obtained from patients with GC or *Helicobacter pylori*-infected gastric cells. However, generating tumors in SCID mice is difficult even 3–4 months after xenotransplantation. Thus, we used organoids derived from gastric adenocarcinoma cells, not those from induced pluripotent stem cells, to evaluate the drug efficacies in the present study.

Using human GC organoids, we compared the effects of antioxidant drugs, including PEA and CA, which are used as specific drugs, as well as SFN, which is the most popularly used drug as phase II enzyme inducing ligand. The growth trend of organoids after PEA and CA treatments was opposite to that after SFN treatment, which stimulated the proliferation and growth of organoids (Figure 2). PEA and CA treatments reduced the growth by 72–87% compared with that in the untreated organoids. We also compared the invasive activity of the respective organoids using the Transwell^®^ migration and invasion assays (Figure 3). Both PEA and CA treatments decreased the invasive activities by approximately 40–55% compared with that in the control organoids. By contrast, SFN treatment enhanced the invasive activity by approximately 1.8-fold to 2.0-fold compared with that in the control organoids. These findings are consistent with those of the xenotransplantation assays (Figure 4a). The PEA and CA treatments significantly reduced the tumor weights and sizes by 15–35% compared with those in the case of control organoids. By contrast, SFN treatment caused a 3.0–3.5-fold increase in tumor sizes. The immunohistochemical analysis revealed confirmative staining patterns among the samples (Figure 4b). On analyzing the staining efficiency using antibodies against NRF2 in the tumor xenografts, Nrf2 staining was significantly reduced in tumors treated with PEA and CA but increased in tumors treated with SFN. These findings were confirmed by the Western blotting of these organoid-derived tumor samples. Interestingly, the expression of NRF2, JDP2, and TP53 was decreased in tumors derived from the organoids treated with PEA and CA. However, SFN treatment increased the expression of NRF2 and TP53 but not AHR and JDP2 (Figure 4c). This finding indicated a correlation of NRF2 and TP53 expression with tumor development. The expression of JDP2 was also consistent. The correlation of NRF2 and TP53 expression with the development of cancer cells and drug resistance has been reported [44,45]. In addition, we previously demonstrated that Jdp2 was involved with Nrf2 in the regulation of antioxidation response genes [46]. Moreover, p53 also controlled the Jdp2-dependent regulation [47]. Thus, further experiments to clarify the regulation mediated by JDP2, NRF2, and TP53 will be required to understand the development of GC.

We next examined ROS generation and caspase 3/7 activities (Figure 5) and found that both were significantly decreased in the organoids that were treated with PEA and CA. These findings were consistent with the expression of NRF2 and TP53 proteins (Figure 4c). Moreover, NRF2 knockdown reversed the effects of PEA, CA, and SFN treatments on the invasive activities of the organoids (Figure 6).

Taken together, these studies of organoids treated with PEA and CA further validate our previous findings using 2D mouse embryonic fibroblast cells to induce ROS production and AHR promoter activation.

Here, we also found that the effects of PEA and CA on human GC development were inconsistent with those of SFN. This unexpected finding demonstrates that although PEA or CA and SFN are antioxidant drugs, they affect tumor development differently.

SFN is known to induce S-phase cell cycle arrest and apoptosis in a TP53-dependent manner in GC 2D cells, such as BGC-823 and BGC-803 cells [25]. It was also reported to inhibit GC stem cells via repressing the sonic hedgehog axis [48] and other reports showed that SFN inhibited the progression of GC cells [49]. However, the effects of SFN are rather complex. In contrast to the case in cancer cells, it was reported that SFN acted pro-oxidatively in primary human T cells. It increased the intracellular ROS levels in T cells and decreased the glutathione levels, thereby inhibiting T cell activation and effector functions [50]. This finding indicates a double-edged sword.

Another important aspect is the differential regulation of AHR and NRF2 transcription factor-mediated response. PEA and CA activate the NRF2-mediated phase II response but inhibit the AHR-mediated responses. However, SFN committed the activation of phase II enzyme genes with NRF2-mediated ARE and AHR-mediated phase II response gene like NQO1 [27,28]. This difference might be cancer-type or drug-specific. Previously, Masutani et al. [51] reported that PEA was a novel thioredoxin inducer. In fact, PEA-induced ARE-activation was suppressed by overexpressing wild-type KEAP1, whereas SFN-induced activation seemed to be partially suppressed. Thus, PEA may activate the KEAP1/NRF2 system more than SFN. The exact reason why these compounds exhibited opposing effects on GC development as described here should be addressed further in the future.

The results with Nutlin3 treatment [51] indicated that the TP53 gene was mutated (Figure 4d). Exposure of SFN to 3D-GC organoids promoted the cancer development, but treatment with either PEA or CA resulted in cancer regression (Figure 4a,b). They showed the cancer development contradiction. In general, it is accepted that expression of mutant TP53 involved the gain of oncogenic-specific activities accentuating the malignant phenotype, biochemical alterations like proliferation, and changes of DNA repair machinery. These effects culminate in the acquisition of drug resistance and cancer recurrence often seen in cancer stem cells (CSCs) expressing mutant TP53 [52]. In addition, we found that the expressions of TP53 and NRF2 were significantly reduced in treatment with PEA or CA but increased in SFN exposure. Thus, in TP53 mutated cancer cells or the 3D-GC organoids, the treatment with either PEA or CA differs from that of SFN.

The expression of JDP2 was also reversed between treatment with PEA/CA and SFN (Figure 4c). In HCT116 p53^−/−^ cells, we found that the wild-type TP53 repressed the Jdp2 promoter activity, but mutated TP53 did not [47]. We also reported that Jdp2 stimulated the Nrf2-dependent antioxidant response [46]. Therefore, we speculate that the expressions of TP53 and JDP2 might be coregulated with each other and affected the NRF2 expression in TP53 mutant cancer cells or organoids. Thus, further studies are required to clarify these molecular relationships of mutated TP53, JDP2, and NRF2.

In the case of a phase II clinical trial, only minor effects of SFN were noted in patients with prostate cancer [53]. Furthermore, no effects of SFN were reported in patients with breast cancer [54]. In general, several reports showed that SFN inhibited the progression of GC in 2D-based cancer cells [25,48,49,55,56]. Thus, we need to study further to clarify these inconsistent results in 2D cells and 3D organoids at the molecular levels. Then, we can find new therapeutic approaches to solve this problem in clinical application to overcome these difficulties in the near future. Using the human 3D organoid system, we found that SFN enhanced GC development. By contrast, the effects of PEA and CA were completely different in our studies. In addition, both PEA and CA did not produce in vivo mutagenicity when administered at doses of up to 1000 mg/kg/day in mice [57]. In summary, we demonstrated that treatment with either PEA or CA had effects that were contradictory to those of SFN on human GC. Moreover, because 3D organoids also contain cancer stem cells and stem cell niches, the findings obtained using 3D organoids differed from those obtained using 2D cells. Thus, using 3D organoids is a powerful approach to investigating cancer stem cells and stem cell niches.

## 4. Materials and Methods

### 4.1. Animals

This study followed the animal welfare guidelines for the care and use of laboratory animals by the Animal Care Committee of the National Laboratory Animal Center (NLAC (TN)-111-W-014; Control Breeding Project (Kazushige Yokoyama) 9 March 2022–31 December 2024), Taiwan; and Kaohsiung Medical University, Taiwan (108244, Gastric intestinal organoids as 3D model for studies of cancer stem cells, their niches and drug development, 1 August 2020–31 July 2023).

### 4.2. Organoid Preparation and Chemicals

The gastric organoids were prepared from the human metastatic gastric adenocarcinoma organoid line, HCM-BROD-0045-C16 (PDM-116™; American Type Culture Collection, Manassas, VA, USA). This organoid line was cultivated using the recommended protocol with certain modifications [29,30]. Chemicals used for the experiments included PEA ([*S*]-4-[1-methylethenyl]-1-cyclohexene-1-carboxaldehyde; P0866; Tokyo Chemical Industry Co., Ltd., Tokyo, Japan), CA ([2*E*]-3-phenylprop-2-enal; W228613; (Merck, Darmstadt, Germany), L-SFN (S6317; Sigma-Aldrich, St Louis, MO, USA), and dimethyl sulfoxide (DMSO; D2650; Sigma-Aldrich). The gastric organoids were incubated with antioxidant drugs or DMSO as a control for indicated time periods, respectively. The viability was measured using the CellTiter-Glo^®^ 3D Cell Viability Assay (G9683; Promega Corp., Madison, WI, USA) using the recommended protocol. The gastric organoid morphologies were recorded and scanned using the Cell3iMager neo scanner (CC-3000; SCREEN Holdings Co., Ltd., Kyoto, Japan).

### 4.3. Bioreactor Culture

The culture method used in the present study was adopted from the simple bioreactor-based method described by Przepiorski et al. [32] with a slight modification for gastric organoids. The gastric organoids were prepared for culture in a 5 mL or 30 mL disposable ABLE Biott bioreactor (#BWV-S005A or S03A; REPROCELL Inc., Yokohama, Japan) on a magnetic stir plate, and they resembled embryoid body structures that are naturally formed for 3D organoid suspension cultures in the bioreactor.

### 4.4. Immunohistochemistry

The sample slides were rehydrated in 10 mM phosphate buffered saline (PBS), and the antigens were retrieved in a sodium citrate buffer (10 mM sodium citrate and 0.05% Tween 20, pH 6.0) at 121 °C for 15 min. Endogenous peroxidase within the sample sections was terminated by treatment with 3% H_2_O_2_ in PBS for 15 min, followed by rinsing with PBS plus 0.1% Triton X-100. After washing and blocking of sample slides, they were reacted with antibodies against α-smooth muscle actin (α-SMA; 1:50; #E14344; Spring Bioscience Corporation, Pleasanton, CA, USA), integrin α6 (1:50; #MAB13501; R&D systems, Minneapolis, MN, USA), NRF2 (1:50; #BS1258; Bioworld Technology, Cambridge, UK), AHR (1:50; sc-8088; Santa Cruz Biotechnology, Dallas, TX, USA), or p21^Cip1^ (1:50; sc-397; Santa Cruz Biotechnology) for 2 h, and then incubated with anti-mouse or rabbit horseradish peroxidase (HRP)-conjugated antibodies (BioTnA, Kaohsiung, Taiwan) for 30 min. After washing thoroughly, the chromogen was developed using 3,3′-diaminobenzidine staining, and the samples were counterstained using hematoxylin (HE). The slides were rinsed using water and covered with a resin-based mounting medium (BioTnA) after dehydration. Histopathology analysis was performed at the Experimental Animal Division, RIKEN BioResource Research Center, Japan, and the National Laboratory Animal Center, Taiwan. Tissue-Hematoxylin (HE) stain was detected using an Olympus CKX41 microscope (Olympus, Tokyo, Japan), and all images were scanned and then saved using a TissueFAXS microscope (Tissue Gnostics, Vienna, Austria). The images were quantified using the open-access Fiji/ImageJ analysis software (version 2.1.1) Download https://imagej.net/software/), accessed on 7 June 2012 [57].

### 4.5. Preparation of shNRF2 and shRNA-Mediated Knockdown

Short hairpin RNA (shRNA) lentivirus against human *NRF2* (TRCN0000007555,TRCN0000007558, and TRCN00000273494) was obtained from the siRNA core center at Academia Sinica (Taipei, Taiwan). The shRNA was introduced into gastric organoids at a multiplicity of infection of 10 or 100. After 24 h, fresh culture medium containing 10% fetal calf serum was added. To confirm the knockdown efficiency of the shRNA, the cells were harvested at 48 h after shRNA infection and analyzed using immunoblotting as described elsewhere [46,58].

### 4.6. Measurement of Cellular ROS Levels

The ROS-Glo™ H_2_O_2_ assay kit (Promega Co., Madison, WI, USA) was used to measure the intracellular ROS levels. After 2 h of treatment with antioxidant drugs and H_2_O_2_, the cells from organoids were washed twice using Hank’s balanced salt solution (GIBCO-Thermo Fisher Scientific, Waltham, MA, USA) and then incubated with ROS-Glo™ Detection Solution for 20 min. The luminescence was quantified using a GloMax^®^ luminometer (Promega).

### 4.7. Measurement of Viable Cells

The CellTiter-Glo^®^ 3D Cell Viability Assay (Promega) was performed according to the manufacturer’s instructions to measure the number of viable cells in 3D cell cultures based on the quantification of adenosine triphosphate, which is a marker for the presence of metabolically active cells. This assay is formulated with a more robust lytic capacity and designed for use with microtissues produced in 3D cell cultures.

### 4.8. Measurement of Caspase-Glo 3/7 Activities

The activities of caspases 3 and 7 were quantified using the Caspase-Glo^®^ 3/7 assay kits (Promega). The relative luminescence units were calculated using a GloMax^®^ luminometer (Promega).

### 4.9. Invasion Assay

The gastric organoids were transferred into a Transwell^®^ plate with an 8.0 μm pore polycarbonate membrane insert (#3422; Corning, NY, USA) coated with Matrigel (Corning; 1 mg/mL) without serum. The Transwell was then put on a plate containing organoid culture medium with DMSO or antioxidant drugs for 7 days. The migrated cells on the lower surface of the membrane were fixed, stained, and counted using a microscope according to the manufacturer’s instructions.

### 4.10. Xenograft Injection

Moreover, 3D organoids were cultured in medium as recommended at a density of 1 × 10^6^ cells in a 10 cm dish and transfected with the required overexpression or shutdown vectors using the lentivirus system 48 h before injection [29]. The injection medium was combined with Matrigel Matrix (Corning, Glendale, AZ, USA) as recommended: 1 × 10^5^ cells were injected subcutaneously into severe deficient immunocompetency (SCID) mice; the xenografts were traced every week to three months after the injection; the xenograft weights were measured in milligrams; and the xenografts were fixed in 4% formaldehyde for biopsy analysis.

### 4.11. Western Blotting

After the culture of the organoids, the radioimmunoprecipitation assay buffer (#20-188; Millipore-Merck, Darmstadt, Germany) was used to extract the total protein. Aliquots of protein lysates (20 μg) were separated using NuPAGE™ 4–12% Bis-Tris protein gels in 1.5 mm, 15-well dishes (NP0335BOX; Thermo Fisher Scientific, Waltham, MA, USA), then transferred onto 0.45 μm Immobilon^®^-P transfer polyvinylidene difluoride membranes (IPVH00010; Merck) for 1 h at 100 V (fixed) at 10 °C using a TE22 transfer system (Hoefer Inc., Holliston, MA, USA). Blots were stained using Ponceau S (P17170; Merck) to monitor the amount of the transferred proteins. The polyvinylidene difluoride membranes were then probed using primary antibodies against NRF2 (1:1000; sc-722; Santa Cruz Biotechnology, Santa Cruz, CA, USA), AHR (1:1000; sc-8088; Santa Cruz Biotechnology), TP53 (1:1000; #2524; Cell Signaling Technology, Danvers, MA, USA;), p21^Cip1^ (1:1000; sc-397; Santa Cruz Biotechnology), Jun dimerization protein 2 (Jdp2; 1:1000; a gift from Dr. Aronheim), and β-ACTIN (1:1000; sc-47778; Santa Cruz Biotechnology), followed by the following secondary antibodies: anti-rabbit immunoglobulin G HRP-conjugated antibody (1:3000; #7074; Cell Signaling Technology, Danvers, MA, USA) and anti-mouse immunoglobulin G HRP-conjugated antibody (1:3000; #7076; Cell Signaling Technology). The results were analyzed using a ChemiDoc XRS+ instrument (Bio-Rad, Hercules, CA, USA).

### 4.12. Statistics

Data are presented as the mean ± standard error of the mean (SEM). Statistical comparisons between experimental conditions were conducted using GraphPad Prism 5.0 (GraphPad Software, San Diego, CA, USA). For multiple comparisons, a one-way analysis of variance (ANOVA) followed by Tukey’s test was performed using GraphPad Prism 7.0 (GraphPad). An unpaired, two-tailed Student’s *t*-test was conducted to compare the control and treatment groups. The Mann–Whitney nonparametric median statistical test was used for cell area analysis. All differences were designated as statistically significant at *p* < 0.05.

## 5. Conclusions

The growth assays of 3D–GC organoid systems and their invasion assays were simple and useful approaches for evaluating the progression of gastric cancers. These methods are suitable for high-throughput drug screening and development of the new drugs for precision medicine. PEA and CA prohibited the tumor progression of GC organoids; by contrast, SFN treatment increased the tumorigenesis. In addition, ROS levels and the NRF2–TP53 axis might be crucial for these screening platforms.

## Figures and Tables

**Figure 1 ijms-24-15911-f001:**
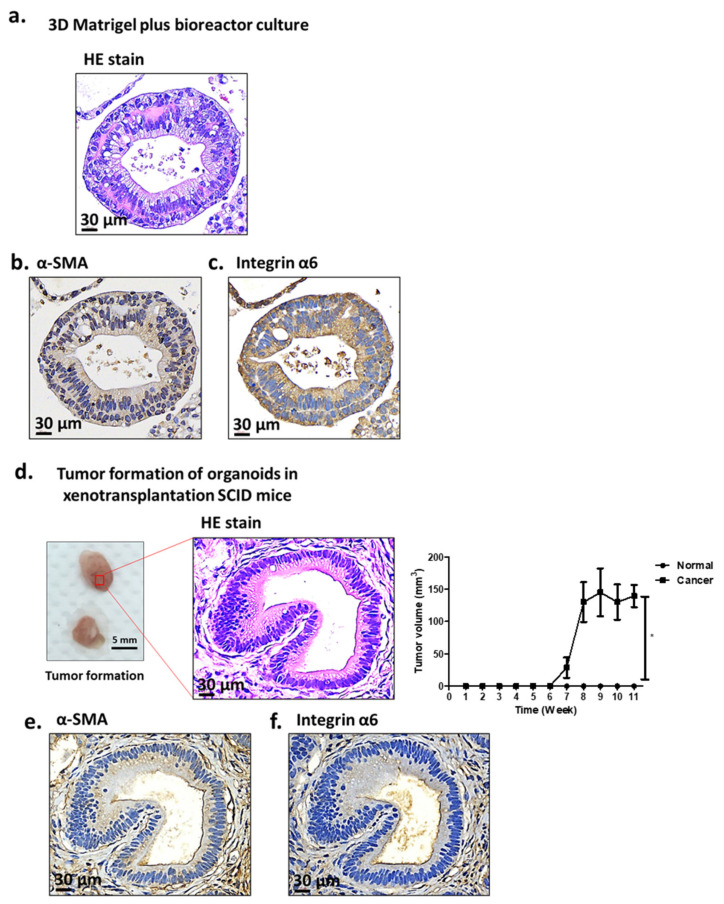
The structure of the antrum organoids after culture in the bioreactor. (**a**–**c**) The organoids were prepared using a bioreactor as described in the Materials and Methods. The tumors are stained by hematoxylin ((**a**): HE stains). The organoids were immune-stained using an anti-α-SMA antibody (**b**) and an anti-integrin α6 antibody (**c**). We found that the organoids contained a central cavity with outer multilayered structures. (**d**–**f**), (**d**) Xenograft tumors on the SCID mice were photographed and HE stains of these tumors were performed ((**d**): HE stains). The tumor volumes from normal gastric organoids and tumor organoids were measured every week and compared until 12 weeks after inoculation, respectively. Data are mean ± SD (*n* = 6). * *p* < 0.05. Tumor formation of the organoids on SCID mice was analyzed using antibodies against α-SMA (panel (**e**)) and integrin α6 (panel (**f**)). Representative images. Scale bars: 30 μm and 5 mm.

**Figure 2 ijms-24-15911-f002:**
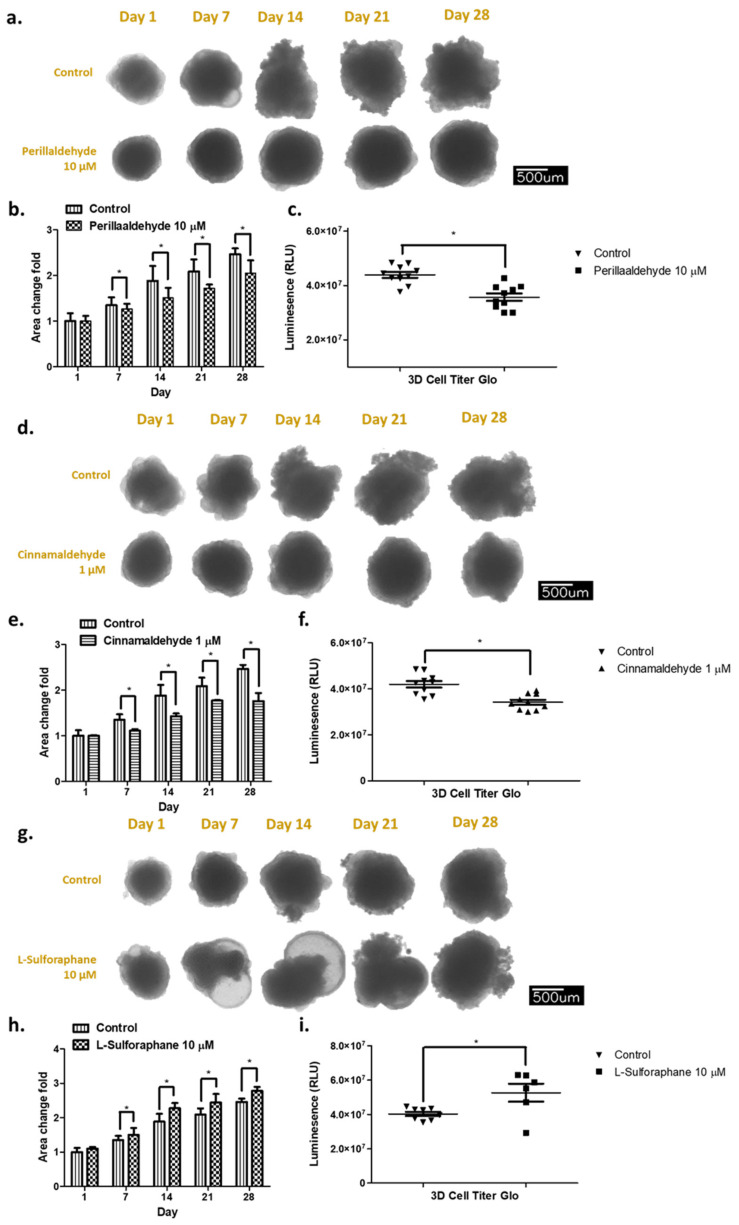
The effects of indicated antioxidant drugs on the morphologies and growth of the gastric organoids after 1, 7, 14, 21, and 28 days of treatment (**a**,**d**,**g**). The bright images of the gastric organoids after exposure to 10 μM PEA, 1 μM CA, and 10 μM SFN, respectively. (**b**,**f**,**i**) The size area of each organoid was quantified as described in the Materials and Methods. The size area of the organoids on control day 1 was considered 1.0 and used to calculate the relative size of each organoid. The area change folds were calculated using growth images. (**c**,**f**,**i**) The viability assay for organoids treated with 10 μM PEA, 1 μM CA, and 10 μM SFN for 7 days was performed, and the luminescence activity was evaluated as described in the Materials and Methods. Data represent the mean ± SEM (*n* = 10 in (**b**,**c**,**g**,**h**); *n* = 6 in (**h**,**i**) and were analyzed using two-way ANOVA with the Tukey post hoc test (**b**,**e**,**h**) and unpaired, two-tailed Student’s *t*-test (**c**,**f**,**i**) (* *p* < 0.05).

**Figure 3 ijms-24-15911-f003:**
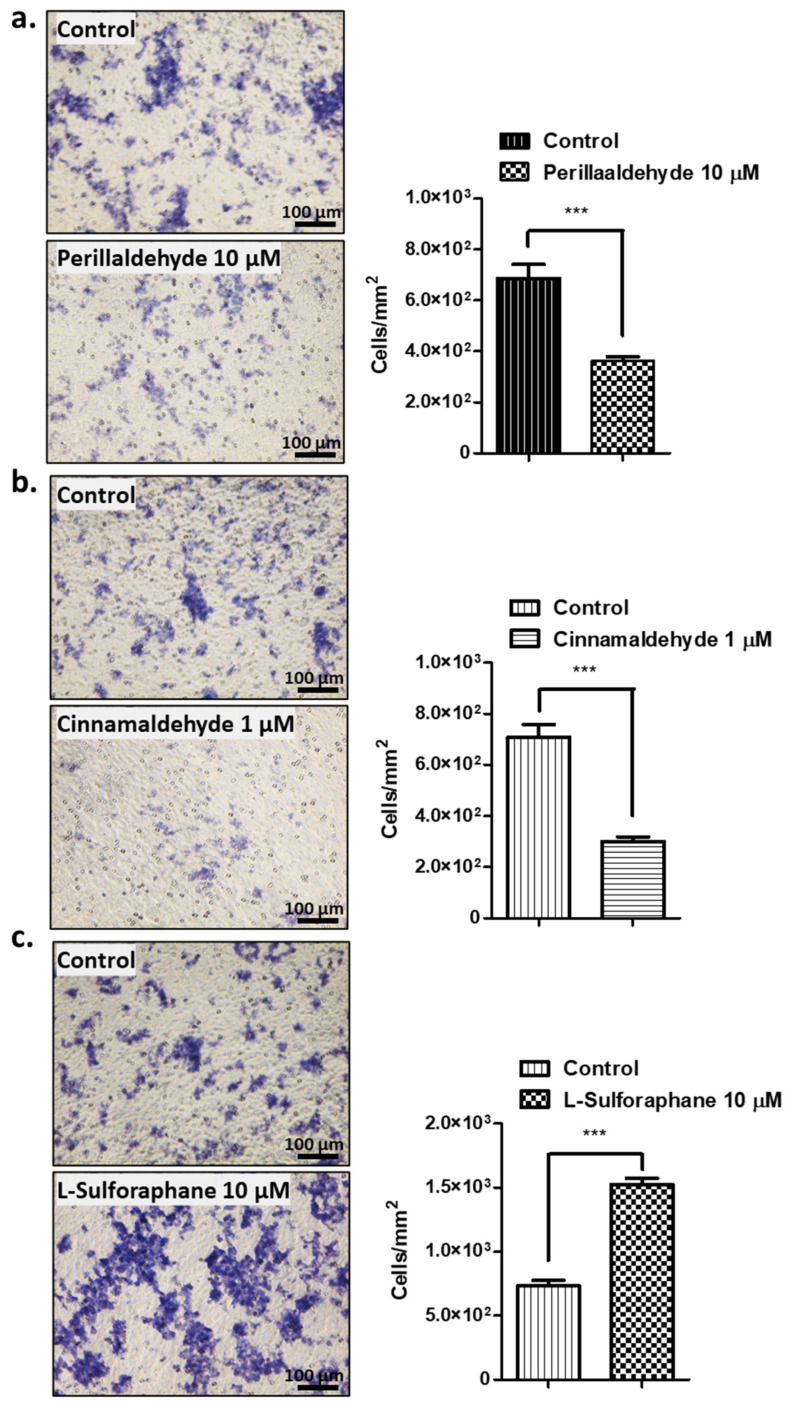
Invasion assays for the organoids after treatment with antioxidant drugs, including PEA (10 μM), CA (1 μM), and SFN (10 μM). (**a**–**c**) The stained migrated cells were detected and quantified as described in the Materials and Methods. Data represent the mean ± SEM (*n* = 6) and were analyzed using unpaired, two-tailed Student’s *t*-test (*** *p* < 0.001). Scale bars: 100 μm. Bars represent the average of triplicates with standard deviation.

**Figure 4 ijms-24-15911-f004:**
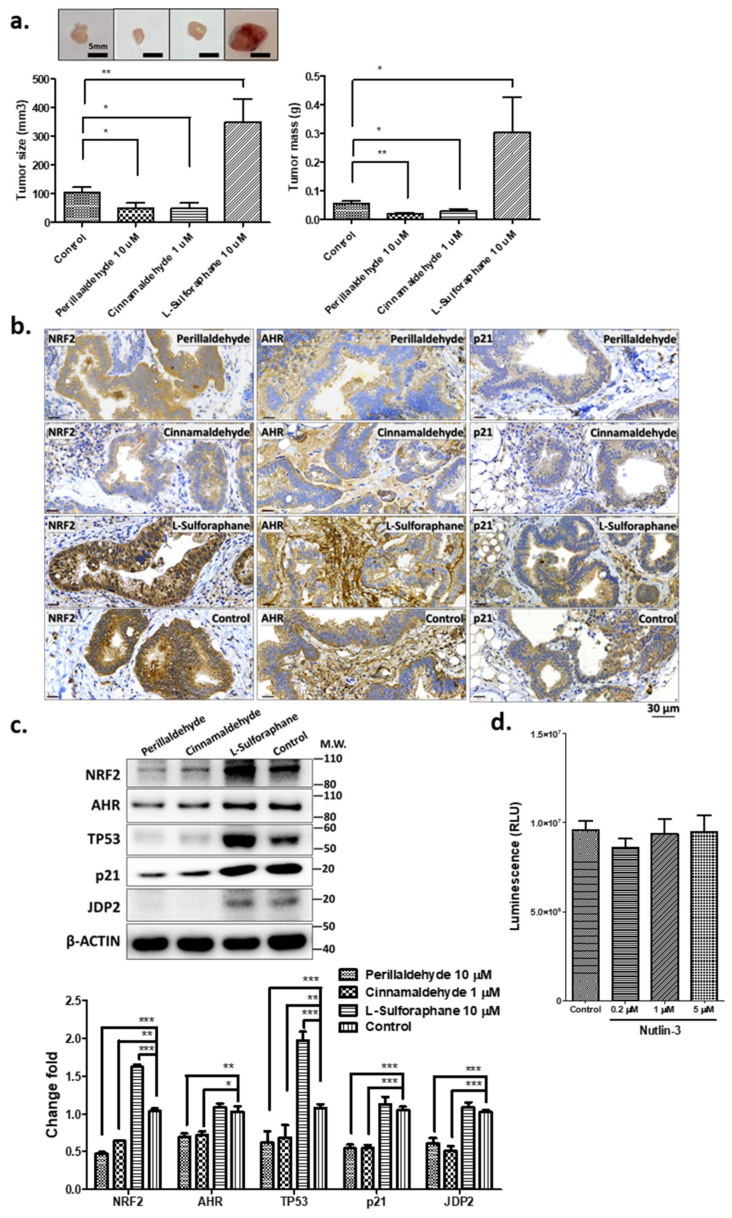
Tumor formation assays for the organoids after treatment with antioxidant drugs, including PEA (10 μM), CA (1 μM), and SFN (10 μM). (**a**) Xenografts after organoid injection of 10^6^ cells in the left/right flanks of SCID male mice. Tumors were harvested 360 days after injection. The tumor sizes and masses were measured. Data represent the mean ± SEM (*n* = 5) and were analyzed using one-way ANOVA with the Tukey post hoc test (* *p* < 0.05; ** *p* < 0.01). (**b**) Immunohistochemistry of tumors derived from organoids treated with PEA (10 μM), CA (1 μM), and SFN (10 μM) in SCID mice using antibodies against NRF2, AHR, and p21^Cip1^. Scale bars: 30 μm. (**c**) Western blot analysis of the expression of NRF2, AHR, TP53, p21^Cip1^, JDP2, and β-ACTIN after treatment with the antioxidant drugs (upper panel (The uncropped data was shown in Appendix A)). Relative expression levels were calculated based on β-ACTIN levels and the control cells with DMSO solvent (0.01%) as 1.0 (lower panel). Data represent the mean ± SEM (*n* = 3). (* *p* < 0.05; ** *p* < 0.01; *** *p* < 0.005). The molecular weight markers are shown on the right side of the gel. (**d**) Effects of nutlin-3 on cell viability of organoids treated with 0.2, 1.0, and 2.0 μM nutlin-3 for 1 week as described elsewhere [40]. The cell viability was estimated using CellTiter-Glo^®^ 3D cell viability assay. Bars represent the average of triplicates with standard deviation.

**Figure 5 ijms-24-15911-f005:**
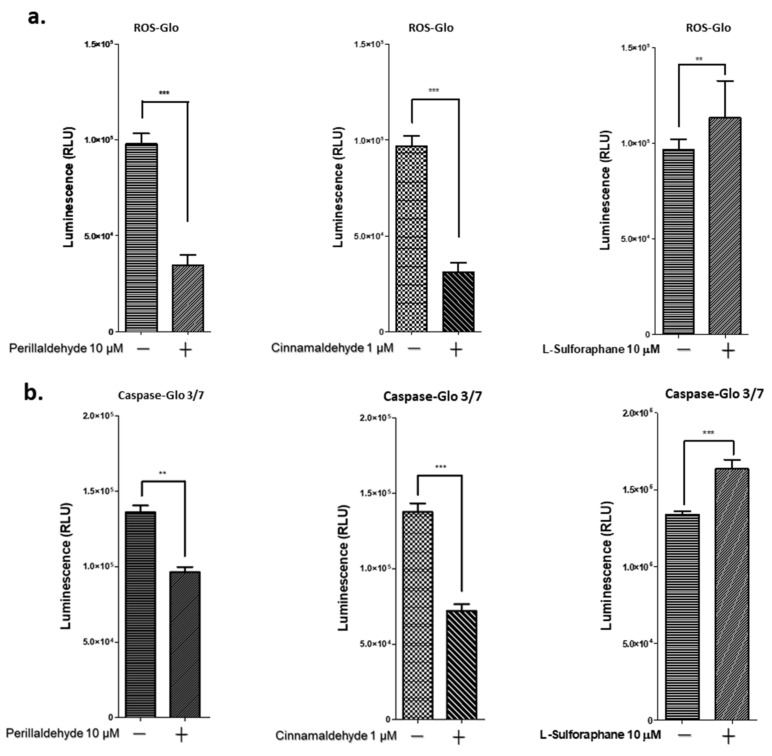
ROS and caspase 3/7 assays for the gastric organoids after antioxidant drug treatment. (**a**) The gastric organoids were treated with 10 μM PEA, 1 μM CA, and 10 μM SFN, respectively, and with 20 μM menadione as a control, and each luminescence activity was measured. The ROS-Glo activity was measured as described in the Materials and Methods. The values of control 20 μM menadione were shown as 1.84 × 10^5^ ± 0.05 (RLU). (**b**) The caspase 3/7 activities were also measured. The values of control 20 μM menadione were shown as 1.63 × 10^5^ ± 0.08 (RLU). + and − mean the presence or absence of indicated drugs. Data represent the mean ± SEM (*n* = 3) and were analyzed using unpaired, two-tailed Student’s *t*-test (** *p* < 0.01; *** *p* < 0.001).

**Figure 6 ijms-24-15911-f006:**
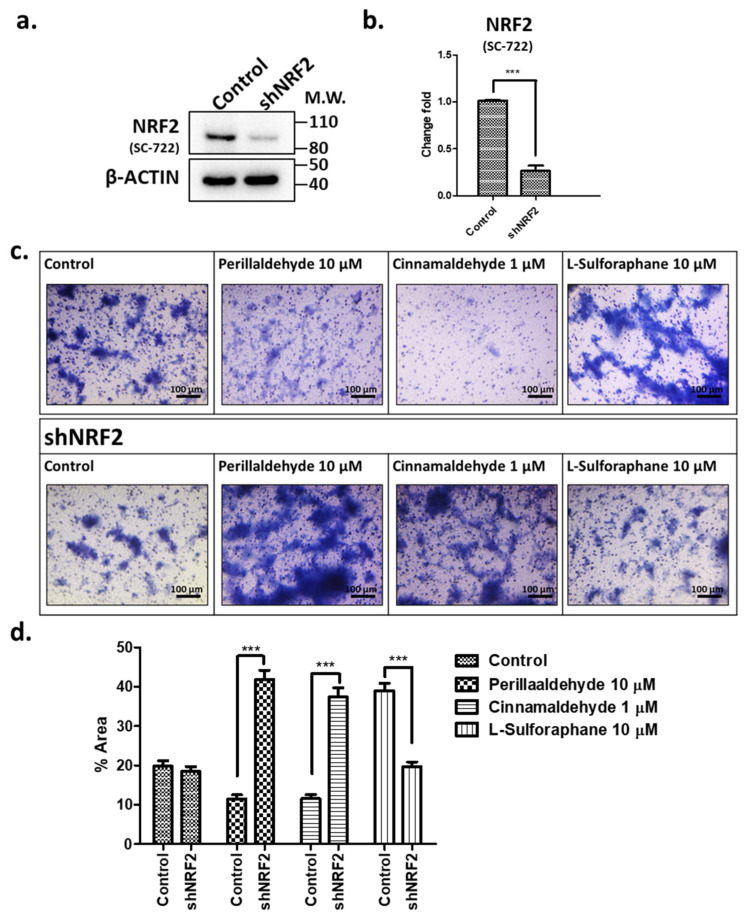
Effects of *shNRF2* on invasive activities of the gastric organoids treated with antioxidant drugs. (**a**,**b**) Expression of NRF2 in organoids treated with shRNA after 10 μM PEA, 1 μM CA, and 10 μM SFN treatments were examined using Western blotting as described in the Materials and Methods (The uncropped data was shown in Appendix A). (**c**,**d**) Invasion assays for the *shNRF2*-treated organoids after treatment with antioxidant drugs, including PEA (10 μM), CA (1 μM), and SFN (10 μM). The stained migrated cells were detected and quantified as described in the Materials and Methods. Data represent the mean ± SEM (*n* = 6) and were analyzed using one-way ANOVA with the Tukey post hoc test (*** *p* < 0.001). Scale bars: 100 μm.

## Data Availability

The data and materials that support the findings of this study are available from the corresponding author upon reasonable request.

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
