# Peer review of "Heterogeneity of Phase II Enzyme Ligands on Controlling the Progression of Human Gastric Cancer Organoids as Stem Cell Therapy Model"

_ijms, 2023, doi:10.3390/ijms242115911_

Round 1
Reviewer 1 Report
Comments and Suggestions for Authors
Author Response
[Reviewer 1]
The authors described the heterogeneity of phase II enzyme ligands in regulating human gastric cancer organoids as a model for stem cell therapy. Briefly, SFN therapy promoted carcinogenesis, but PEA and CA prevented the growth of tumors in GC organoids. Furthermore, ROS concentrations and the Nrf2-p53 axis may be essential for these screening platforms. However, the study is interesting, and the authors should address the following comments:
- In figure 1d, the authors show tumors from xenotransplanted mice. The addition of tumor volume by the authors would help the manuscript.
(Reply) Thank you so much for your kind suggestion. As you suggested, we have added the data of tumor volume in each transplanted mice after measure every week until 12 weeks in Figure 1d.
- How authors select antioxidant dosages like PEA, CA, and SFN. Explain.
(Reply) Thank you so much for your kind suggestion. These three antioxidants are grouped differently to focus upon the activation of AhR gene (phase I response) and Nrf2 gene (phase II response). Both PEA and CA inhibited the response of AhR signaling but activated Nrf2 pathways, which were published in refs. 20 and 24. The optimal dose of these antioxidation reagents is followed to the references. They reported that the optimal doses for PEA and CA are 1-100 μM and 1-5 μM for keratinocyte cell line. The optimal dose of SFN 5-10 μM is also followed to the reference which was determined by the gastric cancer cell lines (ref. 25). Then, we examined the growth of 3D organoids in the presence of these dosages as described in 3.2 section of Results as well as the reagents of BHQ, N-acetylcysteine, and DMSO. Then, we selected the defined doses of PRA, CA and SFN. Then, we decided the better dose of each antioxidant reagents as PEA 10μM, Ca 1μM and SFN 10μM.
- Include the antioxidant concentration (mg/kg?) and the route of administration (IP or oral) for mice.
(Reply)
Thank you so much for your suggestion. In our experiments we prepared 3D organoids incubated each antioxidant and then injected these 3D organoids into the epidermis, not to treat the mice with these reagents IP or oral.
Here we have used the standard xenotransplantation protocol using the tumor organoids. The organoids were incubated for indicated concentration of PFA (10 μM) and CA (1 μM) and SFN (10 μM) in each culture flask for one week and then, they were inoculated onto the skin on each mice epidermis. After one or two months, the tumor formation was detected and measured the sizer of each tumor. We added these procedures to the Materials and Methods section.
- Figure 3a–c. The authors demonstrated that the organoids' invasive area was decreased by PEA and CA therapy. In contrast, Figure 6c shows that the treatment with antioxidants causes an increase in the shNrf2 organoids' invasive areas. Explain. (Treatment with PEA and CE may cause further reductions in shNrf2 organoids?)
(Reply)
The main finding of this paper is to describe the heterogeneity of antioxidant phase II drugs to control the cancer development through the transcription factor Nrf2. In general, PEA and CA activated the Nrf2 mediated phase II response and then inhibited or no effect of AhR phase I response. However, SFN activated mainly the Nrf2 response of phase II enzyme to induce the antioxidation response, not phase I AhR mediation. SNF has known to inhibit the cancer development and invasion, apoptosis of cancer stem cells (Front Oncol 13, 1089115, 2023). However, this dogma seems to be changed now. SFN was known to activate both AhR and Nrf2 pathways (Free Rad Biol Med 143, 331-140, 2019, Food Sci Nutr 2023, 11, 2277-2287). Thus, the effects of PEA, CA, and SFN exhibited differently against AhR and Nrf2 mediated response. Therefore, we wonder these antioxidation reagents might function differently to control antioxidation response in 3D organoids.
In addition, SFN can modulate the activity of some epigenetic factors, such as histone deacetylases (HDAC), thus impacting the expression of genes involved in tumor initiation and progression (Drug Des Devel Ther 12, 2905-2913, 2018). However, this effect is not seen in ductal carcinoma in situ and no changes were observed in benign tumors or invasive ductal carcinomas (Cancer Prev Res (Phila) (2015) 8:1184–1191; Public Health Nutr (2015) 19:1288–1295). Thus, the effects of SFN as the anticancer stem cell drug are dependent upon the cancer types. In Figure 3c, we found that SFN enhanced the expression of Nrf2 in cancer organoids compared with the untreated cancer organoids. Surprisingly, the expression of p53 was increased by treatment with SFN, and the result of Nutlin3 -treatment indicated that p53 were mutated (Figure 4d). These treatment with SFN promoted the gastric cancer development (Figure 4a and b). We also found the tumor development of p53 mutated gastric cancer organoids was enhanced. The mutation of p53 might contribute to the development of cancer stem cells (CSCs). The drug resistance and cancer recurrence were also mediated by CSCs which expressed the mutant p53 (Carcinogenesis 35, 1196-1208, 2014). Thus, p53-mutation is critical for cancer development.
By contrast, we demonstrated that PEA and CA reduced the cancer development and invasion activities of respective gastric cancer organoids (Figure 4a) and found that Nrf2 expression was significantly decreased (Figure 4c). In addition, the expressions of p53 and Jdp2 were significantly repressed, Thus, in the case of p53 mutated cancer cells, the phase II drugs like PEA or CA showed differently from that of SFN. In this case, we wonder why the phase II ligands such as PEA and CA did not induce the expression of Nrf2 (Figure 4c), which might be the case of p53 mutation. Therefore, we next focused on this variable expression of Nrf2 between PEA or CA and SFN. Using shRNA-Nrf2, we found that the Nrf2 played a critical role for tumor invasion (Figure 6) because the treatment of shRNA-Nrf2 reversed the functions of invasion activities completely. Thus, we now know that Nrf2 expression is one of the direct changes in these phase II drug treatments.
Although we do not know the exact mechanism of this heterogeneity of the phase II drugs, we propose that Nrf2 expression might be a key target for future experiments in p53 mutated cancer cells organoid experiments. The new transcription factor Jdp2 is also expressed variably between PEA or CA and SFN treatment. In HCT116 p53-/- cells, we found that the wild type p53 repressed the Jdp2 promoter activity, but the mutated p53 did not (Xu et al. Biochem Biophys Res Communi 450, 1531-1536, 2014). In addition, we reported that Jdp2 regulated the Nrf2 dependent antioxidant response in the phase II ligands (Tanigawa et al., Cell Death Dis. 5(7): e1344, 2014). Therefore, we speculate that in p53 mutant derived cells, both p53 and Jdp2 might be coregulated each other. Thus, these mechanistic studies will be clarified by further experiments. Thus. In Figure 4c, both expressions of Jdp2 and Nrf2 might be coordinated regulated in mutant p53. We need further experiments to seek for this molecular relationship. We added to describe this speculation in Discussion section to let the audience understand easily.
[Reviewer 1]
The authors described the heterogeneity of phase II enzyme ligands in regulating human gastric cancer organoids as a model for stem cell therapy. Briefly, SFN therapy promoted carcinogenesis, but PEA and CA prevented the growth of tumors in GC organoids. Furthermore, ROS concentrations and the Nrf2-p53 axis may be essential for these screening platforms. However, the study is interesting, and the authors should address the following comments:
- In figure 1d, the authors show tumors from xenotransplanted mice. The addition of tumor volume by the authors would help the manuscript.
(Reply) Thank you so much for your kind suggestion. As you suggested, we have added the data of tumor volume in each transplanted mice after measure every week until 12 weeks in Figure 1d.
- How authors select antioxidant dosages like PEA, CA, and SFN. Explain.
(Reply) Thank you so much for your kind suggestion. These three antioxidants are grouped differently to focus upon the activation of AhR gene (phase I response) and Nrf2 gene (phase II response). Both PEA and CA inhibited the response of AhR signaling but activated Nrf2 pathways, which were published in refs. 20 and 24. The optimal dose of these antioxidation reagents is followed to the references. They reported that the optimal doses for PEA and CA are 1-100 μM and 1-5 μM for keratinocyte cell line. The optimal dose of SFN 5-10 μM is also followed to the reference which was determined by the gastric cancer cell lines (ref. 25). Then, we examined the growth of 3D organoids in the presence of these dosages as described in 3.2 section of Results as well as the reagents of BHQ, N-acetylcysteine, and DMSO. Then, we selected the defined doses of PRA, CA and SFN. Then, we decided the better dose of each antioxidant reagents as PEA 10μM, Ca 1μM and SFN 10μM.
- Include the antioxidant concentration (mg/kg?) and the route of administration (IP or oral) for mice.
(Reply)
Thank you so much for your suggestion. In our experiments we prepared 3D organoids incubated each antioxidant and then injected these 3D organoids into the epidermis, not to treat the mice with these reagents IP or oral.
Here we have used the standard xenotransplantation protocol using the tumor organoids. The organoids were incubated for indicated concentration of PFA (10 μM) and CA (1 μM) and SFN (10 μM) in each culture flask for one week and then, they were inoculated onto the skin on each mice epidermis. After one or two months, the tumor formation was detected and measured the sizer of each tumor. We added these procedures to the Materials and Methods section.
- Figure 3a–c. The authors demonstrated that the organoids' invasive area was decreased by PEA and CA therapy. In contrast, Figure 6c shows that the treatment with antioxidants causes an increase in the shNrf2 organoids' invasive areas. Explain. (Treatment with PEA and CE may cause further reductions in shNrf2 organoids?)
(Reply)
The main finding of this paper is to describe the heterogeneity of antioxidant phase II drugs to control the cancer development through the transcription factor Nrf2. In general, PEA and CA activated the Nrf2 mediated phase II response and then inhibited or no effect of AhR phase I response. However, SFN activated mainly the Nrf2 response of phase II enzyme to induce the antioxidation response, not phase I AhR mediation. SNF has known to inhibit the cancer development and invasion, apoptosis of cancer stem cells (Front Oncol 13, 1089115, 2023). However, this dogma seems to be changed now. SFN was known to activate both AhR and Nrf2 pathways (Free Rad Biol Med 143, 331-140, 2019, Food Sci Nutr 2023, 11, 2277-2287). Thus, the effects of PEA, CA, and SFN exhibited differently against AhR and Nrf2 mediated response. Therefore, we wonder these antioxidation reagents might function differently to control antioxidation response in 3D organoids.
In addition, SFN can modulate the activity of some epigenetic factors, such as histone deacetylases (HDAC), thus impacting the expression of genes involved in tumor initiation and progression (Drug Des Devel Ther 12, 2905-2913, 2018). However, this effect is not seen in ductal carcinoma in situ and no changes were observed in benign tumors or invasive ductal carcinomas (Cancer Prev Res (Phila) (2015) 8:1184–1191; Public Health Nutr (2015) 19:1288–1295). Thus, the effects of SFN as the anticancer stem cell drug are dependent upon the cancer types. In Figure 3c, we found that SFN enhanced the expression of Nrf2 in cancer organoids compared with the untreated cancer organoids. Surprisingly, the expression of p53 was increased by treatment with SFN, and the result of Nutlin3 -treatment indicated that p53 were mutated (Figure 4d). These treatment with SFN promoted the gastric cancer development (Figure 4a and b). We also found the tumor development of p53 mutated gastric cancer organoids was enhanced. The mutation of p53 might contribute to the development of cancer stem cells (CSCs). The drug resistance and cancer recurrence were also mediated by CSCs which expressed the mutant p53 (Carcinogenesis 35, 1196-1208, 2014). Thus, p53-mutation is critical for cancer development.
By contrast, we demonstrated that PEA and CA reduced the cancer development and invasion activities of respective gastric cancer organoids (Figure 4a) and found that Nrf2 expression was significantly decreased (Figure 4c). In addition, the expressions of p53 and Jdp2 were significantly repressed, Thus, in the case of p53 mutated cancer cells, the phase II drugs like PEA or CA showed differently from that of SFN. In this case, we wonder why the phase II ligands such as PEA and CA did not induce the expression of Nrf2 (Figure 4c), which might be the case of p53 mutation. Therefore, we next focused on this variable expression of Nrf2 between PEA or CA and SFN. Using shRNA-Nrf2, we found that the Nrf2 played a critical role for tumor invasion (Figure 6) because the treatment of shRNA-Nrf2 reversed the functions of invasion activities completely. Thus, we now know that Nrf2 expression is one of the direct changes in these phase II drug treatments.
Although we do not know the exact mechanism of this heterogeneity of the phase II drugs, we propose that Nrf2 expression might be a key target for future experiments in p53 mutated cancer cells organoid experiments. The new transcription factor Jdp2 is also expressed variably between PEA or CA and SFN treatment. In HCT116 p53-/- cells, we found that the wild type p53 repressed the Jdp2 promoter activity, but the mutated p53 did not (Xu et al. Biochem Biophys Res Communi 450, 1531-1536, 2014). In addition, we reported that Jdp2 regulated the Nrf2 dependent antioxidant response in the phase II ligands (Tanigawa et al., Cell Death Dis. 5(7): e1344, 2014). Therefore, we speculate that in p53 mutant derived cells, both p53 and Jdp2 might be coregulated each other. Thus, these mechanistic studies will be clarified by further experiments. Thus. In Figure 4c, both expressions of Jdp2 and Nrf2 might be coordinated regulated in mutant p53. We need further experiments to seek for this molecular relationship. We added to describe this speculation in Discussion section to let the audience understand easily.
[Reviewer 1]
The authors described the heterogeneity of phase II enzyme ligands in regulating human gastric cancer organoids as a model for stem cell therapy. Briefly, SFN therapy promoted carcinogenesis, but PEA and CA prevented the growth of tumors in GC organoids. Furthermore, ROS concentrations and the Nrf2-p53 axis may be essential for these screening platforms. However, the study is interesting, and the authors should address the following comments:
- In figure 1d, the authors show tumors from xenotransplanted mice. The addition of tumor volume by the authors would help the manuscript.
(Reply) Thank you so much for your kind suggestion. As you suggested, we have added the data of tumor volume in each transplanted mice after measure every week until 12 weeks in Figure 1d.
- How authors select antioxidant dosages like PEA, CA, and SFN. Explain.
(Reply) Thank you so much for your kind suggestion. These three antioxidants are grouped differently to focus upon the activation of AhR gene (phase I response) and Nrf2 gene (phase II response). Both PEA and CA inhibited the response of AhR signaling but activated Nrf2 pathways, which were published in refs. 20 and 24. The optimal dose of these antioxidation reagents is followed to the references. They reported that the optimal doses for PEA and CA are 1-100 μM and 1-5 μM for keratinocyte cell line. The optimal dose of SFN 5-10 μM is also followed to the reference which was determined by the gastric cancer cell lines (ref. 25). Then, we examined the growth of 3D organoids in the presence of these dosages as described in 3.2 section of Results as well as the reagents of BHQ, N-acetylcysteine, and DMSO. Then, we selected the defined doses of PRA, CA and SFN. Then, we decided the better dose of each antioxidant reagents as PEA 10μM, Ca 1μM and SFN 10μM.
- Include the antioxidant concentration (mg/kg?) and the route of administration (IP or oral) for mice.
(Reply)
Thank you so much for your suggestion. In our experiments we prepared 3D organoids incubated each antioxidant and then injected these 3D organoids into the epidermis, not to treat the mice with these reagents IP or oral.
Here we have used the standard xenotransplantation protocol using the tumor organoids. The organoids were incubated for indicated concentration of PFA (10 μM) and CA (1 μM) and SFN (10 μM) in each culture flask for one week and then, they were inoculated onto the skin on each mice epidermis. After one or two months, the tumor formation was detected and measured the sizer of each tumor. We added these procedures to the Materials and Methods section.
- Figure 3a–c. The authors demonstrated that the organoids' invasive area was decreased by PEA and CA therapy. In contrast, Figure 6c shows that the treatment with antioxidants causes an increase in the shNrf2 organoids' invasive areas. Explain. (Treatment with PEA and CE may cause further reductions in shNrf2 organoids?)
(Reply)
The main finding of this paper is to describe the heterogeneity of antioxidant phase II drugs to control the cancer development through the transcription factor Nrf2. In general, PEA and CA activated the Nrf2 mediated phase II response and then inhibited or no effect of AhR phase I response. However, SFN activated mainly the Nrf2 response of phase II enzyme to induce the antioxidation response, not phase I AhR mediation. SNF has known to inhibit the cancer development and invasion, apoptosis of cancer stem cells (Front Oncol 13, 1089115, 2023). However, this dogma seems to be changed now. SFN was known to activate both AhR and Nrf2 pathways (Free Rad Biol Med 143, 331-140, 2019, Food Sci Nutr 2023, 11, 2277-2287). Thus, the effects of PEA, CA, and SFN exhibited differently against AhR and Nrf2 mediated response. Therefore, we wonder these antioxidation reagents might function differently to control antioxidation response in 3D organoids.
In addition, SFN can modulate the activity of some epigenetic factors, such as histone deacetylases (HDAC), thus impacting the expression of genes involved in tumor initiation and progression (Drug Des Devel Ther 12, 2905-2913, 2018). However, this effect is not seen in ductal carcinoma in situ and no changes were observed in benign tumors or invasive ductal carcinomas (Cancer Prev Res (Phila) (2015) 8:1184–1191; Public Health Nutr (2015) 19:1288–1295). Thus, the effects of SFN as the anticancer stem cell drug are dependent upon the cancer types. In Figure 3c, we found that SFN enhanced the expression of Nrf2 in cancer organoids compared with the untreated cancer organoids. Surprisingly, the expression of p53 was increased by treatment with SFN, and the result of Nutlin3 -treatment indicated that p53 were mutated (Figure 4d). These treatment with SFN promoted the gastric cancer development (Figure 4a and b). We also found the tumor development of p53 mutated gastric cancer organoids was enhanced. The mutation of p53 might contribute to the development of cancer stem cells (CSCs). The drug resistance and cancer recurrence were also mediated by CSCs which expressed the mutant p53 (Carcinogenesis 35, 1196-1208, 2014). Thus, p53-mutation is critical for cancer development.
By contrast, we demonstrated that PEA and CA reduced the cancer development and invasion activities of respective gastric cancer organoids (Figure 4a) and found that Nrf2 expression was significantly decreased (Figure 4c). In addition, the expressions of p53 and Jdp2 were significantly repressed, Thus, in the case of p53 mutated cancer cells, the phase II drugs like PEA or CA showed differently from that of SFN. In this case, we wonder why the phase II ligands such as PEA and CA did not induce the expression of Nrf2 (Figure 4c), which might be the case of p53 mutation. Therefore, we next focused on this variable expression of Nrf2 between PEA or CA and SFN. Using shRNA-Nrf2, we found that the Nrf2 played a critical role for tumor invasion (Figure 6) because the treatment of shRNA-Nrf2 reversed the functions of invasion activities completely. Thus, we now know that Nrf2 expression is one of the direct changes in these phase II drug treatments.
Although we do not know the exact mechanism of this heterogeneity of the phase II drugs, we propose that Nrf2 expression might be a key target for future experiments in p53 mutated cancer cells organoid experiments. The new transcription factor Jdp2 is also expressed variably between PEA or CA and SFN treatment. In HCT116 p53-/- cells, we found that the wild type p53 repressed the Jdp2 promoter activity, but the mutated p53 did not (Xu et al. Biochem Biophys Res Communi 450, 1531-1536, 2014). In addition, we reported that Jdp2 regulated the Nrf2 dependent antioxidant response in the phase II ligands (Tanigawa et al., Cell Death Dis. 5(7): e1344, 2014). Therefore, we speculate that in p53 mutant derived cells, both p53 and Jdp2 might be coregulated each other. Thus, these mechanistic studies will be clarified by further experiments. Thus. In Figure 4c, both expressions of Jdp2 and Nrf2 might be coordinated regulated in mutant p53. We need further experiments to seek for this molecular relationship. We added to describe this speculation in Discussion section to let the audience understand easily.
[Reviewer 1]
The authors described the heterogeneity of phase II enzyme ligands in regulating human gastric cancer organoids as a model for stem cell therapy. Briefly, SFN therapy promoted carcinogenesis, but PEA and CA prevented the growth of tumors in GC organoids. Furthermore, ROS concentrations and the Nrf2-p53 axis may be essential for these screening platforms. However, the study is interesting, and the authors should address the following comments:
- In figure 1d, the authors show tumors from xenotransplanted mice. The addition of tumor volume by the authors would help the manuscript.
(Reply) Thank you so much for your kind suggestion. As you suggested, we have added the data of tumor volume in each transplanted mice after measure every week until 12 weeks in Figure 1d.
- How authors select antioxidant dosages like PEA, CA, and SFN. Explain.
(Reply) Thank you so much for your kind suggestion. These three antioxidants are grouped differently to focus upon the activation of AhR gene (phase I response) and Nrf2 gene (phase II response). Both PEA and CA inhibited the response of AhR signaling but activated Nrf2 pathways, which were published in refs. 20 and 24. The optimal dose of these antioxidation reagents is followed to the references. They reported that the optimal doses for PEA and CA are 1-100 μM and 1-5 μM for keratinocyte cell line. The optimal dose of SFN 5-10 μM is also followed to the reference which was determined by the gastric cancer cell lines (ref. 25). Then, we examined the growth of 3D organoids in the presence of these dosages as described in 3.2 section of Results as well as the reagents of BHQ, N-acetylcysteine, and DMSO. Then, we selected the defined doses of PRA, CA and SFN. Then, we decided the better dose of each antioxidant reagents as PEA 10μM, Ca 1μM and SFN 10μM.
- Include the antioxidant concentration (mg/kg?) and the route of administration (IP or oral) for mice.
(Reply)
Thank you so much for your suggestion. In our experiments we prepared 3D organoids incubated each antioxidant and then injected these 3D organoids into the epidermis, not to treat the mice with these reagents IP or oral.
Here we have used the standard xenotransplantation protocol using the tumor organoids. The organoids were incubated for indicated concentration of PFA (10 μM) and CA (1 μM) and SFN (10 μM) in each culture flask for one week and then, they were inoculated onto the skin on each mice epidermis. After one or two months, the tumor formation was detected and measured the sizer of each tumor. We added these procedures to the Materials and Methods section.
- Figure 3a–c. The authors demonstrated that the organoids' invasive area was decreased by PEA and CA therapy. In contrast, Figure 6c shows that the treatment with antioxidants causes an increase in the shNrf2 organoids' invasive areas. Explain. (Treatment with PEA and CE may cause further reductions in shNrf2 organoids?)
(Reply)
The main finding of this paper is to describe the heterogeneity of antioxidant phase II drugs to control the cancer development through the transcription factor Nrf2. In general, PEA and CA activated the Nrf2 mediated phase II response and then inhibited or no effect of AhR phase I response. However, SFN activated mainly the Nrf2 response of phase II enzyme to induce the antioxidation response, not phase I AhR mediation. SNF has known to inhibit the cancer development and invasion, apoptosis of cancer stem cells (Front Oncol 13, 1089115, 2023). However, this dogma seems to be changed now. SFN was known to activate both AhR and Nrf2 pathways (Free Rad Biol Med 143, 331-140, 2019, Food Sci Nutr 2023, 11, 2277-2287). Thus, the effects of PEA, CA, and SFN exhibited differently against AhR and Nrf2 mediated response. Therefore, we wonder these antioxidation reagents might function differently to control antioxidation response in 3D organoids.
In addition, SFN can modulate the activity of some epigenetic factors, such as histone deacetylases (HDAC), thus impacting the expression of genes involved in tumor initiation and progression (Drug Des Devel Ther 12, 2905-2913, 2018). However, this effect is not seen in ductal carcinoma in situ and no changes were observed in benign tumors or invasive ductal carcinomas (Cancer Prev Res (Phila) (2015) 8:1184–1191; Public Health Nutr (2015) 19:1288–1295). Thus, the effects of SFN as the anticancer stem cell drug are dependent upon the cancer types. In Figure 3c, we found that SFN enhanced the expression of Nrf2 in cancer organoids compared with the untreated cancer organoids. Surprisingly, the expression of p53 was increased by treatment with SFN, and the result of Nutlin3 -treatment indicated that p53 were mutated (Figure 4d). These treatment with SFN promoted the gastric cancer development (Figure 4a and b). We also found the tumor development of p53 mutated gastric cancer organoids was enhanced. The mutation of p53 might contribute to the development of cancer stem cells (CSCs). The drug resistance and cancer recurrence were also mediated by CSCs which expressed the mutant p53 (Carcinogenesis 35, 1196-1208, 2014). Thus, p53-mutation is critical for cancer development.
By contrast, we demonstrated that PEA and CA reduced the cancer development and invasion activities of respective gastric cancer organoids (Figure 4a) and found that Nrf2 expression was significantly decreased (Figure 4c). In addition, the expressions of p53 and Jdp2 were significantly repressed, Thus, in the case of p53 mutated cancer cells, the phase II drugs like PEA or CA showed differently from that of SFN. In this case, we wonder why the phase II ligands such as PEA and CA did not induce the expression of Nrf2 (Figure 4c), which might be the case of p53 mutation. Therefore, we next focused on this variable expression of Nrf2 between PEA or CA and SFN. Using shRNA-Nrf2, we found that the Nrf2 played a critical role for tumor invasion (Figure 6) because the treatment of shRNA-Nrf2 reversed the functions of invasion activities completely. Thus, we now know that Nrf2 expression is one of the direct changes in these phase II drug treatments.
Although we do not know the exact mechanism of this heterogeneity of the phase II drugs, we propose that Nrf2 expression might be a key target for future experiments in p53 mutated cancer cells organoid experiments. The new transcription factor Jdp2 is also expressed variably between PEA or CA and SFN treatment. In HCT116 p53-/- cells, we found that the wild type p53 repressed the Jdp2 promoter activity, but the mutated p53 did not (Xu et al. Biochem Biophys Res Communi 450, 1531-1536, 2014). In addition, we reported that Jdp2 regulated the Nrf2 dependent antioxidant response in the phase II ligands (Tanigawa et al., Cell Death Dis. 5(7): e1344, 2014). Therefore, we speculate that in p53 mutant derived cells, both p53 and Jdp2 might be coregulated each other. Thus, these mechanistic studies will be clarified by further experiments. Thus. In Figure 4c, both expressions of Jdp2 and Nrf2 might be coordinated regulated in mutant p53. We need further experiments to seek for this molecular relationship. We added to describe this speculation in Discussion section to let the audience understand easily.

Reviewer 2 Report
Comments and Suggestions for Authors
The authors have examined the effects of three anti oxidant compounds on growth and characteristics of gastric organoids.
The different effects of the compounds are interesting and highlight the need for further investigation of these compounds for the utilisation in cancer treatment.
A couple of minor corrections are required:
- the bars on the graphs should be made to be more different between groups in all figures.
- line 274; this 75-80% decrease in growth is not reflected in the graph 2b. Looks far less. Please clarify.
- figure 3 looks horizontally stretched and out of proportion.
Author Response
Reviewer 2;
The authors have examined the effects of three anti oxidant compounds on growth and characteristics of gastric organoids. The different effects of the compounds are interesting and highlight the need for further investigation of these compounds for the utilisation in cancer treatment. A couple of minor corrections are required:
-the bars on the graphs should be made to be more different between groups in all figures.
(Reply)
According to reviewer 2, we have reorganized the bars on all figures to make clear to show the differences.
- line 274; this 75-80% decrease in growth is not reflected in the graph 2b. Looks far less. Please clarify.
(Reply)
As suggested by the reviewer, we have recalculated the decreasing extents and reevaluated. PEA treatment reduced growth by 74-87% and CA treatment decreased by 72-85%. We changed this description as suggested by the reviewer.
- figure 3 looks horizontally stretched and out of proportion.
(Reply)
We have revised the horizontal bars in Figure 3.

Reviewer 3 Report
Comments and Suggestions for Authors
11) In their investigation, the authors assessed the anti-tumor potential of Perillaldehyde, Cinnamic acid, and L-Sulforaphane against gastric cancer organoids. Their findings revealed that Perillaldehyde (PEA) and Cinnamic acid (CA) exhibited significant antiproliferative effects, while L-Sulforaphane (SFN) did not demonstrate a substantial impact. This is noteworthy because previous literature has consistently indicated the antiproliferative and antioxidant properties of SFN. However, the current study's results deviate from this established pattern, which challenges the consistency of prior reports.
12) The authors have encountered unexpected results in the current study, where the anticipated anticancer effects, such as the activation of ROS production, induction of apoptosis, and a significant increase in Caspase-3 activity, were not observed, contrary to numerous prior studies. In light of this, the authors are now faced with the task of validating these unexpected findings.
13) In Figure 4, the labeling of the change fold graph is missing, and it is not explained in the figure legend. Please review and revise the manuscript to include this essential information.
14) Figure 5 presents data on ROS and Caspase-3/7 activity with the use of the drug menadione for treatment. However, the manuscript lacks a discussion of this treatment in both the results and methods sections. Additionally, it's important to ensure uniform formatting of the figures. Please address these issues for clarity and consistency.
15) The symbols "+" and "–" in Figure 4 lack a clear explanation. Please provide their meanings in the legend section for clarity.
Comments on the Quality of English LanguageModerate editing of English language required
Author Response
Reviewer 3.
- In their investigation, the authors assessed the anti-tumor potential of Perillaldehyde, Cinnamic acid, and L-Sulforaphane against gastric cancer organoids. Their findings revealed that Perillaldehyde (PEA) and Cinnamic acid (CA) exhibited significant antiproliferative effects, while L-Sulforaphane (SFN) did not demonstrate a substantial impact. This is noteworthy because previous literature has consistently indicated the antiproliferative and antioxidant properties of SFN. However, the current study's results deviate from this established pattern, which challenges the consistency of prior reports.
(Reply)
The paper demonstrated the heterogeneity of antioxidant phase II drugs to control the cancer development of 3D cancer organoids through the transcription factor Nrf2. In general, PEA and CA activated the Nrf2 mediated phase II response and then inhibited AhR phase I response (refs. 20 and 24). However, SFN activated mainly the Nrf2 response of phase II enzyme to induce the antioxidation response, not phase I AhR mediation. SFN then inhibited the cancer development and invasion, apoptosis of cancer stem cells (Front Oncol 13, 1089115, 2023). However, this dogma seems to be changed now. SFN was known to activate both AhR and Nrf2 pathways (Free Rad Biol Med 143, 331-140, 2019, Food Sci Nutr 2023, 11, 2277-2287) and functioned through the balance of AhR/Nrf2. Thus, the effects of PEA, CA, and SFN exhibited differently against AhR and Nrf2 mediated response. Therefore, we wonder these antioxidation reagents might function differently to control antioxidation response in 3D organoids.
In addition, SFN can modulate the activity of some epigenetic factors, such as histone deacetylases (HDAC), thus impacting the expression of genes involved in tumor initiation and progression (Drug Des Devel Ther 12, 2905-2913, 2018). However, this effect is not seen in ductal carcinoma in situ and no changes were observed in benign tumors or invasive ductal carcinomas (Cancer Prev Res (Phila) (2015) 8:1184–1191; Public Health Nutr (2015) 19:1288–1295). Thus, the effects of SFN as the anticancer stem cell drug are dependent upon the cancer types. In Figure 4c, we found that SFN enhanced the expression of Nrf2 in cancer organoids compared with the untreated cancer organoids. Surprisingly, the expression of p53 was increased by treatment with SFN, and the result of Nutlin3-treatment indicated that p53 were mutated (Figure 4d). These treatment with SFN promoted the gastric cancer development (Figure 4a and b). We also found the tumor development of p53 mutated gastric cancer organoids was enhanced. The mutation of p53 might contribute to the development of cancer stem cells (CSCs). The drug resistance and cancer recurrence were also mediated by CSCs which expressed the mutant p53 (Carcinogenesis 35, 1196-1208, 2014). Thus, p53-mutation is critical for cancer development.
By contrast, we demonstrated that PEA and CA reduced the cancer development and invasion activities of respective gastric cancer organoids (Figure 4a and b) and found that Nrf2 expression was significantly decreased (Figure 4c). In addition, the expressions of p53 and Jdp2 were significantly repressed, Thus, in the case of p53 mutated cancer cells, the phase II drugs like PEA or CA exhibited differently from that of SFN. In this case, we wonder why the phase II ligands such as PEA and CA did not induce the expression of Nrf2 (Figure 4c), which might be the case of p53 mutation. Therefore, we next focused on this variable expression of Nrf2 between PEA or CA and SFN. Using shRNA-Nrf2, we found that the Nrf2 played a critical role for tumor invasion (Figure 6) because the treatment of shRNA-Nrf2 reversed the functions of invasion activities completely. Thus, we now know that Nrf2 expression is one of the direct changes in these phase II drug treatments.
Although we do not know the exact mechanism of this heterogeneity of the phase II drugs, we propose that Nrf2 expression might be a key target for future experiments in p53 mutated cancer cells organoid experiments. The new transcription factor Jdp2 is also expressed variably between PEA or CA and SFN treatment. In HCT116 p53-/- cells, we previously found that the wild type p53 repressed the Jdp2 promoter activity, but the mutated p53 did not (ref. 49). In addition, we reported that Jdp2 regulated the Nrf2 dependent antioxidant response in the phase II ligands (Ref. 34). Therefore, we speculate that in p53 mutant derived cells, both p53 and Jdp2 might be coregulated each other. Thus, these mechanistic studies will be clarified by further experiments. Thus. In Figure 4c, both expressions of Jdp2 and Nrf2 might be coordinated regulated in mutant p53. We need further experiments to seek for this molecular relationship. We added to describe this speculation in Discussion section.
- The authors have encountered unexpected results in the current study, where the anticipated anticancer effects, such as the activation of ROS production, induction of apoptosis, and a significant increase in Caspase-3 activity, were not observed, contrary to numerous prior studies. In light of this, the authors are now faced with the task of validating these unexpected findings.
(Reply)
As shown in the above answer to the question 1 raised by the reviewer. This finding of SFN treatment as reversed as compared with 2D cancer cells and their xenotransplantation (Front Oncol 13, 1089115, 2023). The main reason is the Nrf2-p53-Jdp2 axis to control the antioxidation reaction. However, mutated p53 was increased and then Nrf2 expression is also enhanced in the case of SFN treatment. Thus, we have to seek for the interaction between mutated p53 and Nrf2 and increased the ROS, caspase 3 and then develop the cancer progression. By contrast, these events were completely shut down in cells treated with PEA and CA. These Nrf2-mutant 53 or Nrf2-Jdp2-mutant p53 axis should be clarified in future why we observed differently using the 3D organoid, not 2D cells. This possible speculation is added in the Discussion section.
- In Figure 4, the labeling of the change fold graph is missing, and it is not explained in the figure legend. Please review and revise the manuscript to include this essential information.
(Reply)
As suggested by the reviewer, we added the labels of each graph of Figure 4c lower panel (see below of Western blotting as upper panel). We also added the sentences as the control DMSO taken as 1.0. in Figure 4c legends.
- Figure 5 presents data on ROS and Caspase-3/7 activity with the use of the drug menadione for treatment. However, the manuscript lacks a discussion of this treatment in both the results and methods sections. Additionally, it's important to ensure uniform formatting of the figures. Please address these issues for clarity and consistency.
(Reply)
As suggested by the reviewer. We added the control activities of menadione as the caspase activity and ROS level in Figure legends.
5) The symbols "+" and "–" in Figure 5 lack a clear explanation. Please provide their meanings in the legend section for clarity.
(Reply)
As suggested by the reviewer. We have added the explanation in the Figure 5 legends.
In addition, we have changed the letter of AhR, Nrf2 and p53 to AHR, NRF2 and TP53 for human nomenclatures in the Text.
Round 2
Reviewer 3 Report
Comments and Suggestions for Authors
Accepted for publication
Comments on the Quality of English LanguageNone